# In Vitro and In Vivo Characterization of a Broadly Neutralizing Anti-SARS-CoV-2 Antibody Isolated from a Semi-Immune Phage Display Library

**DOI:** 10.3390/antib11030057

**Published:** 2022-09-06

**Authors:** Edith González-González, Gregorio Carballo-Uicab, Juana Salinas-Trujano, María I. Cortés-Paniagua, Said Vázquez-Leyva, Luis Vallejo-Castillo, Ivette Mendoza-Salazar, Keyla Gómez-Castellano, Sonia M. Pérez-Tapia, Juan C. Almagro

**Affiliations:** 1Unidad de Desarrollo e Investigación en Bioterapéuticos (UDIBI), Escuela Nacional de Ciencias Biológicas, Instituto Politécnico Nacional, Mexico City 11340, Mexico; 2Laboratorio Nacional para Servicios Especializados de Investigación, Desarrollo e Innovación (I+D+i) para Farmoquímicos y Biotecnológicos, LANSEIDI-FarBiotec-CONACyT, Mexico City 11340, Mexico; 3Departamento de Inmunología, Escuela Nacional de Ciencias Biológicas, Instituto Politécnico Nacional (ENCB-IPN), Mexico City 11340, Mexico; 4GlobalBio, Inc., 320 Concord Ave, Cambridge, MA 02138, USA

**Keywords:** COVID-19, broadly neutralizing antibody, non-clinical efficacy, Delta variant, Omicron variant

## Abstract

Neutralizing antibodies targeting the receptor-binding domain (RBD) of SARS-CoV-2 are among the most promising strategies to prevent and/or treat COVID-19. However, as SARS-CoV-2 has evolved into new variants, most of the neutralizing antibodies authorized by the US FDA and/or EMA to treat COVID-19 have shown reduced efficacy or have failed to neutralize the variants of concern (VOCs), particularly B.1.1.529 (Omicron). Previously, we reported the discovery and characterization of antibodies with high affinity for SARS-CoV-2 RBD Wuhan (WT), B.1.617.2 (Delta), and B.1.1.529 (Omicron) strains. One of the antibodies, called IgG-A7, also blocked the interaction of human angiotensin-converting enzyme 2 (hACE2) with the RBDs of the three strains, suggesting it may be a broadly SARS-CoV-2 neutralizing antibody. Herein, we show that IgG-A7 efficiently neutralizes all the three SARS-CoV-2 strains in plaque reduction neutralization tests (PRNTs). In addition, we demonstrate that IgG-A7 fully protects K18-hACE2 transgenic mice infected with SARS-CoV-2 WT. Taken together, our findings indicate that IgG-A7 could be a suitable candidate for development of antibody-based drugs to treat and/or prevent SARS-CoV-2 VOCs infection.

## 1. Introduction

COVID-19, caused by the severe acute respiratory syndrome coronavirus 2 (SARS-CoV-2), was declared a global pandemic by The World Health Organization (WHO) in March 2020 [1]. Despite significant efforts by the scientific community, healthcare organizations and governments to control the pandemic, over half a billion individuals have been infected worldwide with this viral disease, leading to more than 6 million fatalities since the emergence of COVID-19 in December 2019 [2]. Several vaccines have received an Emergency Use Authorization (EUA) by the Food and Drug Administration (FDA), and over hundred more are in preclinical and clinical developments, or have been approved to be used in diverse countries [3]. However, although vaccines are the forefront biotechnological products for COVID-19 prevention, not all individuals who have received one or several doses of the vaccines develop a protective immune response against SARS-CoV-2, and none of the vaccines are entirely effective, particularly in individuals who have been infected with the variants of concern (VOCs), including Delta (B.1.617.2) and Omicron (B.1.1.529) [4].

In parallel to the development and approval of vaccines, as well as massive vaccination campaigns, nine therapeutic antibodies have received EUA by the FDA and/or European Agency of Medicines (EMA) as prophylactic and therapeutic drugs [5]. In fact, the first antibody-based drug to treat COVID-19, called REGEN-COV (imdevimab plus casirivimab), received FDA EUA on November 2020, before any COVID-19 vaccine reached the market [6,7]. Yet, similarly to the vaccines, most of the EUA therapeutic antibodies have shown limited efficacy or have progressively failed to protect patients infected with the emergence of VOCs [8,9,10]. This, compounded with the fact that SARS-CoV-2 has continued to evolve, generating immune escape variants as shown in the recent emergence of the BA.3, BA.4 and BA.5 sub-lineages of Omicron [11], has spurred a continuous search for new and more efficacious antibodies to treat COVID-19.

In a previous report [12], we described the isolation of antibodies from a semi-immune scFv phage display library built with four synthetic V_L_ libraries combined with the immune V_H_ repertoire of a convalescent COVID-19 patient. The patient received a single dose of Convidecia™ (CanSino Biologics Inc., Tianjin, China) and a few months later was infected with SARS-CoV-2 Delta, exhibiting typical symptoms of COVID-19. After panning the semi-immune libraries with the RBD wild-type (WT), a panel of 30+ anti-SARS-CoV-2 scFvs with diverse functional profiles were obtained. One of the antibodies, called IgG-A7, recognized the RBDs WT, Delta, and Omicron with K_D_ values of 0.68, 0.24, and 8.18 nM, respectively. In addition, IgG-A7 blocked the interaction between the RBD WT and hACE2, suggesting this antibody as a potential candidate for developing broadly neutralizing anti-SARS-CoV-2 prophylactic and/or therapeutic drugs.

In this work, we report that IgG-A7 neutralizes authentic SARS-CoV-2 WT, Delta, and Omicron viral isolates in vitro, with a neutralization potency superior to CB6, the precursor of etesevimab, the second therapeutic COVID-19 antibody that received FDA EUA as a cocktail with bamlanivimab [13]. More importantly, we show that IgG-A7 protects 100% of K18-hACE2 transgenic mice expressing hACE2 infected with SARS-CoV-2 WT at a dose of 5 mg/kg. Taken together, these results make IgG-A7 a promising candidate to develop a therapeutic antibody to treat and/or prevent SARS-CoV-2 infections.

## 2. Materials and Methods

### 2.1. Monoclonal Antibody Production and Purification

IgG-A7 was cloned and expressed as reported in Mendoza-Salazar et al. [12]. To set up and validate the assays, as well as a reference in the characterization of IgG-A7, CB6 was used as a positive control [14]. CB6 is the precursor of etesevimab [15], a therapeutic antibody EUA FDA approved as a cocktail with bamlanivimab to treat COVID-19 [13]. As a negative control, we used D1.3, a well-known anti-lysozyme antibody [16]. CB6 and D1.3 variable regions were cloned with the same isotype than IgG-A7 (hIgG1), as well as expressed and purified following the same procedures than IgG-A7.

### 2.2. Developability Assessment

IgG-A7 was subjected to an analytical platform of validated methodologies to assess its physicochemical profile for pharmaceutical developability and biological activity [17,18,19]. This platform considers determining protein content, purity, identity/integrity, and thermal stability. The tests include Size Exclusion Chromatography (SEC), reducing and non-reducing polyacrylamide gel electrophoresis (SDS-PAGE) and Melting Temperature (Tm) assessment by extrinsic fluorometry using the Protein Thermal Shift™ assay (Thermo Fisher Scientific, Waltham, MA, USA, Cat. No.: 4462263). In addition, the content of endotoxin of the purified sample was determined as quality control for in vitro and in vivo assays by the Limulus amebocyte lysate test (Charles River Laboratory, Wilmington, MA, USA, Cat. No.: R11012).

These analytical techniques were performed as follows. The concentration was determined in The Epoch System (Bio Tek Instruments, Winooski, VT, USA) using the extinction coefficient 1.0 (mg/mL)cm^−1^. SEC was performed on an H class Acquity UPLC system (Waters^®^), and the data were acquired at 280 nm using the Empower™ software from the same vendor. The components of the sample were separated using an Acquity BEH 200 SEC 1.7 µm column (4.6 × 150 mm) (Waters^®^, Milford, MA, USA) at 30 °C and a pH 6.8 with PBS (sodium phosphate/sodium chloride solution; 50 mM/150 mM) running at 0.4 mL/min. SDS-PAGE was performed at 1 μg/lane using Any kD™ TGX Stain-Free™ Protein Gels (Bio-Rad, Hercules, CA, USA, Cat. No: 4568123S) and a Mini-PROTEAN^®^ system (Bio-Rad). Gel images were acquired with a ChemiDoc system (Bio-Rad) and analyzed using Image Lab software (Bio-Rad). A Protein Thermal Shift™ assay was performed using 8 µg/well of IgG-A7 and the changes in the structure were monitored as per the manufacturer’s instructions from 5 to 99 °C on a Fast 7500 Real-Time PCR System (ThermoFisher Scientific). Data were processed with Protein Thermal Shift™ Software (ThermoFisher Scientific) using a multi-Tm detection mode.

### 2.3. Mass Spectrometry Characterization

In addition to the above-mentioned tests, IgG-A7 intact and deglycosylase masses were assessed by mass spectrometry (MS) coupled with Ultra High-Performance Chromatography (UPLC) as described in Perdomo-Abúndez et al. [17]. In brief, IgG-A7 was diluted with 0.01% formic acid (FA) solution (Thermo Fisher Scientific, Cat. No. 85158) at 2 mg/mL for intact mass analysis. For the deglycosylation mass analysis, IgG-A7 was diluted with 0.250 M Tris buffer solution (pH 7.5) at 2 mg/mL and heated at 80 °C for 10 min. Seven μL of PNGase (500,000 U/mL; Roche Cat. No. 11047817001) was added to samples of 150 μL and incubated at 37 °C for 18 h. The deglycosylation reaction was stopped with 10 μL of 20% FA at room temperature, immediately filtered through 0.22 μm centrifugal filters (Millipore, Cat. No. SCGP00525) and placed in glass vials for UPLC-MS injection. Both intact and deglycosylation mass analyses were performed on a Vion mass spectrometer (ESI-IMS-QTof) coupled with a UPLC system (Waters^®^) using an Acquity^®^ UPLC BEH C4 column (1.7 μm, 300 Å, 2.1 mm × 100 mm) and binary mobile phase composed of water 0.1% FA and 0.1% Acetonitrile. Spectrometric data were acquired and processed using the UNIFI software (Waters^®^).

### 2.4. Binding to the Fcγ Receptors and FcRn by Surface Plasmon Resosnance (SPR)

The affinity of IgG-A7 for FcγRIA, FcγRIIA, FcγRIIIA and FcRn was determined in a BIACore T200 (Cytiva). To capture the receptors an anti-His antibody (Capture Kit; Cytiva, Vancouver, BC, Canada, Cat. No.: 28995056) was coupled to a CM5 chip (Cytiva, Vancouver, BC, Canada, Cat. No.: BR100399) using the amine coupling kit, following the manufacturer’s instructions, i.e., the surface was activated using a solution containing 0.2 M N-ethyl- N-dimethyl aminopropylcarbodiimide (EDC) and 50 mM N-hydroxysuccinimide (NHS) for 10 min. The anti-His antibody was diluted to 50 μg/mL in 10 mM sodium acetate (pH 4.5). Recombinant FcRn (R&D System, Minneapolis, MN, USA, Cat. No.: 8639-FC) was directly immobilized on the CM5 at 5 µg/mL in 10 mM sodium acetate, pH 5.0, following the same procedure as the anti-His antibody. The immobilized anti-HIS antibody or FcRn ranged from 9000 to 15,000 RUs.

His-tagged FcγRIA, FcγRIIA and FcγRIIIA (R&D System, Cat. No.: 1257-FC, 1330-CD and 4325-FC, respectively) were injected at 1 μg/mL for 150 s using a flow rate of 10 μL/min in the coated cells with the anti-His antibody (active cells). IgG-A7 was diluted in HBS-EP IX pH 7.4 (Cytiva) and flown over the active cells at a concentration range of 100–12.5 nM, 800–100 nM, and 800–50 nM for FcγRIA, FcγRIIA and FcγRIIIA, respectively. The contact and dissociation times were 300 s/450 s, 25 s/45 s and 300 s/450 s for FcγRIA, FcγRIIA and FcγRIIIA, respectively. The sensor chip surfaces were regenerated using 10 mM Glycine-HCl pH 1.5 (Cytiva, Vancouver, BC, Canada, Cat. No.: BR100354) after each assay cycle. Binding to FcRn was assessed with IgG-A7 in PBS 1X (Cytiva, Vancouver, BC, Canada, Cat. No.: BSS-PBS-1 × 6)/Tween-20 0.05% pH 6.0 as running buffer. A multi-cycle kinetics procedure was applied with a range of concentrations of 300 to 25 nM, with the contact and dissociation times of 180 s/300 s, respectively. After each assay cycle, a 30 s-pulse of PBS at pH 8.0, containing 0.05% Tween-20 was used to regenerate the surfaces. All of the experiments were performed at 25 °C. The data collected from a blank flow cell were subtracted from the active cells and the resulting data fit to a 1:1 model with BIA evaluation software (Cytiva).

### 2.5. SARS-CoV-2 Isolates

SARS-CoV-2 WT, Delta, and Omicron variants were isolated from patients with the typical COVID-19 symptoms and a positive SARS-CoV-2 RT-PCR diagnostic. Nasopharyngeal exudates were carried out under written informed consent following a protocol approved by the research ethics committee of the ENCB (approval number CEI-ENCB-SH-003-2020). Positive nasopharyngeal exudates were stored in Eagle’s Minimum Essential Medium (EMEM) (ATCC, Manassas, VA, USA, Cat. No.: 30-2003). SARS-CoV-2 was propagated in Vero E6 cells (ATCC, Cat. No.: CRL-1586) for 60 h and frozen for one day to lyse the cells. The supernatants were collected and titrated by plaque assay using Vero E6 cells [20].

RNA isolation from culture supernatants was performed using MagMAX Viral and Pathogen Nucleic Acid Isolation Kits (Applied Biosystems™, Thermo Fisher Scientific, Cat. No.: A42352) according to the manufacturer’s instructions. For the whole-genome sequencing of the isolated viruses COVID-19 ARTIC v3 Illumina library construction and sequencing protocol V.4 was used. The genome sequences are deposited in GenBank, accession numbers: OL790194 (WT), OM060237 (Delta) and ON651664 (Omicron).

The sample collection procedure followed the principles established in the Declaration of Helsinki [21]. Isolation and active virus manipulations, as well as efficacy studies were performed at BSL2+ facilities, with strict biosafety standards and risk assessment protocols according to the specifications of the WHO Laboratory Biosafety Manual, Four Edition and the Guidance for General Laboratory Safety Practices during the COVID-19 Pandemic of CDC [22,23,24,25].

### 2.6. Plaque Reduction Neutralization Test (PRNT)

Vero E6 cells were grown in 75 cm^2^ plates with EMEM and 10% Fetal Bovine Serum (FBS) (Gibco, Cat. No.: A4736401) to 80% of confluency. The cell culture was harvested and titrated using trypan blue (Sigma-Aldrich, Darmstadt, Germany, Cat. No.: 93595). A total of 10^5^ cells per well were incubated at 37 °C with 5% CO_2_ in a 24-plate for 20 h to obtain 90% of confluency. SARS-CoV-2 WT, Delta, and Omicron variants were mixed at 70–130 PFU/mL with serial dilutions of IgG-A7, CB6 (positive control), or D1.3 (negative control) and incubated at 37 °C for one hour for WT and Delta variants, and 2 h for Omicron. The mixtures were added to the plated Vero E6 cells and incubated at 37 °C for 1.5 h in 5% CO_2_. The solution was removed and the overlay media (Carboxymethyl cellulose 1% p/v, FBS 1% v/v, EMEM) was added. The plates were incubated for 4 days at 37 °C in 5% CO_2_. Afterwards, 300 µL of formaldehyde 37% were added to each well and incubated overnight at 4 °C to inactivate the viral particles. The inactivated overlay was removed with abundant water and 500 µL of 0.2% methyl violet in 10% formaldehyde solution was added to dye the wells. Lytic plaques in each well were counted and reported as a percentage of neutralization for each dilution.

### 2.7. Efficacy Assay

Groups of male and female mice K18-hACE2 transgenic for the hACE2 (B6.Cg-Tg(K18ACE2)2Prlmn/JHEMI; Jackson Laboratories, strain number 034860, USA) were used in the efficacy studies. Mice were maintained under controlled conditions of temperature, humidity, and noise, with a light/dark cycle of 12/12 h, and fed with commercial chow (LabDiet, St. Louis, MO, USA, Cat. No.: 5010) and water ad libitum. Infection experiments were conducted with SARS-CoV-2 WT—see above. The viral stock was diluted in EMEM to obtain 10^3^ PFU/40 µL and intranasally administered to the mice, previously anesthetized with 65 mg/Kg of Ketamine and 13 mg/Kg of Xylazine. IgG-A7 was given intraperitoneally in a single dose of 5 mg/Kg weight 24 h post infection to a group of six mice (three male and three female). Survival was monitored daily for 14 days. Viral load in the lung was determined for RT-PCR to detect SARS-CoV-2 *E* gene copies on day 14 post-infection. Clinical signs were evaluated daily. Mice were euthanized when showed irreversible signs of disease. All of the animal experiments were performed according to the regulations outlined in the NOM-062-ZOO-1999 [26].

## 3. Results

### 3.1. Expression and Physicochemical Characterization of IgG-A7

The yield of IgG-A7 in HEK 293T cells was 20 mg/L after four days of transfection and Protein A purification. The endotoxin content was <0.125 UE/mL. Figure 1 shows SEC, SDS-PAGE and Thermal stability profiles of IgG-A7. In the SEC experiment no high-molecular weight aggregates were detected, with a monomeric content close to 100%. A single band was seen at 138 kDa in non-reducing conditions. In reducing conditions, two bands of 52 kDa and 25 kDa were observed, typical of IgG heavy and light chains, respectively. Thermal stability analysis indicated that IgG-A7 underwent two main unfolding transitions, one at 68.5 °C and the second at 82.1 °C. The first transition is typical of the C_H_2 unfolding domain of a hIgG1 [27], whereas the second transition should correspond to unfolding of the Fab/C_H_3 domain.

The identity of purified IgG-A7 was assessed by intact and deglycosylated mass analyses. IgG-A7 had five main glycoforms (Figure 2A), which ranged from 147,550 Da to 148,314 Da. Deglycosylated IgG-A7 (Figure 2B) had three main sequence isoforms 144,937, 145,062 and 145,175 Da, which correspond to IgG-A7 with no C-terminal Lysine, IgG-A7 with one C-terminal Lysine (most abundant) and IgG-A7 with two C-terminal Lysines, respectively.

Having assessed the biophysical profile of IgG-A7 with results consistent with those reported for therapeutic antibodies [27], we determined the interaction of IgG-A7 with FcγRIa, FcγRIIa and FcγRIIIa, which mediate effector functions in innate effector cells such as natural killer (NK) cells and macrophages [28]. We also evaluated the interaction with the FcRn, which is involved in modulating the serum half-life of antibodies. Figure 3 shows the BIAcore profiles of the interactions of IgG-A7 with the Fcγ receptors, yielding K_D_ values of 0.02, 0.42 and 0.38 µM for FcγRIa, FcγRIIa and FcγRIIIa, respectively. The K_D_ of IgG-A7 for the FcRn was 2.74 µM. These values matched reported values for diverse hIgG1 therapeutics antibodies (see discussion).

### 3.2. Neutralization Assay In Vitro

To determine the in vitro neutralization potency of IgG-A7 we used PRNT (Figure 4). IgG-A7 neutralized SARS-CoV-2 WT, Delta and Omicron with NC_50_ values of 0.56, 0.06, and 2.92 nM, respectively. In comparison, CB6 NC_50_ values were 2.74 and 0.59 nM for SARS-CoV-2 WT and Delta, respectively. Thus, the neutralization potency of IgG-A7 was five-fold higher than CB6 for SARS-CoV-2 WT and 10-fold superior for SARS-CoV-2 Delta. As expected, CB6 did not neutralize Omicron [29].

### 3.3. Efficacy in Transgenic Mice

To determine whether the neutralizing activity observed in PRNT translated into SARS-CoV-2 protection in vivo, a group of six K18-ACE2 mice (three male and three female) expressing hACE2 were infected with SARS-CoV-2 WT at 1 × 10^3^ PFU and treated with 5 mg/Kg of IgG-A7 24 h post-infection. As a control group, five K18-ACE2 mice (three female and two male) were infected with SARS-CoV-2 WT (1 × 10^3^ PFU) but were not treated with IgG-A7. As second control group of six K18-ACE2 mice (three male and three female) were not infected nor treated with IgG-A7.

Survival of the mice reached 100% protection (Figure 5A) in the group treated with IgG-A7. In contrast, only 20% survived in the SARS-CoV-2 infected group but untreated with IgG-A7. All the uninfected mice survived. Analysis of the viral load in the lungs of treated mice compared to untreated mice (Figure 5B) indicated a highly significant lower viral load in the former than the latter, suggesting that IgG-A7 reduced the replication of SARS-CoV-2 in the lungs of mice.

## 4. Discussion

In the previous sections, we first characterized IgG-A7 with a battery of physicochemical assays including SEC, SDS-PAGE, Thermal stability, intact and deglycosylated masses, as well as binding to Fcγ receptors and FcRn. Table 1 compares the values obtained in this report with those established by Pharma/Biotech companies to define whether an antibody qualifies for further development or needs to be optimized [27]. Although more testing of IgG-A7 needs to be performed to move the antibody to preclinical development, the physicochemical characterization of IgG-A7 matched all of the success criteria for a well-behaved IgG1 and hence, we decided to characterize this promising antibody further in functional assays.

From a functional viewpoint, IgG-A7 neutralized SARS-CoV-2 WT, Delta and Omicron in PRNT. The comparison with CB6 indicated a 5-to-10-fold increase in neutralization potency of IgG-A7 with respect to CB6. Further, IgG-A7 protected, at a dose of 5 mg/Kg, K18-ACE2 mice expressing hACE2 infected with SARS-CoV-2 WT. If we extrapolate this dose to an average human weighing 70 Kg, it will result in a 350 mg dose. As a reference, etesevimab (CB6) is dosed at 1400 mg in a cocktail with 700 mg of bamlanivimab [30]. Thus, the approved therapeutic dose for etesevimab is 4-fold higher than that calculated for IgG-A7. While the calculation of the IgG-A7 potential therapeutic dose is somewhat simplistic, it provides a context to guess how IgG-A7 might be dosed in humans. In fact, a lower dose of IgG-A7 with respect to CB6 is consistent with the higher IgG-A7 in vitro neutralization potency seen in PRNT when compared to CB6. More importantly, IgG-A7 neutralized Omicron in vitro.

**Table 1 antibodies-11-00057-t001:** Summary of IgG-A7 developability profile.

Attribute	Units	IgG-A7	Success Criteria
Expression yield	mg/L	20	12–16
Endotoxin	UE/mL	<0.125	<15
Purity (SEC)	%	100	>95
	kDa	205	~150
Integrity (SDS-PAGE)	Heavy chain (kDa)	51.9	~50
	Light chain (kDa)	25.4	~25
	Whole molecule (kDa)	138.1	~150
Thermal stability	Tm1 (°C)	68.5	68
	Tm2 (°C)	82.1	68–83
Intact Mass (Main peak)	Da [ppm]	147,967	Should correspond to the calculated mass based on the amino acid sequence plus glycans
Deglycosylated (Main peak)	Da [ppm]	145,062	Should correspond to the calculated mass based on the amino acid sequence
Fcγ receptors (reference values are those measured with a similar capture reagent, i.e., anti-HIS antibody)	FcγRI (µM)	0.02	0.0009–0.052 [31]
	FcγRIIA (µM)	0.42	4.20–6.00 [31]
	FcγRIIIA (µM)	0.38	0.089–2.166 [31]
FcRn	µM	2.74	0.9–4.3 [32]

In this regard, nine therapeutic antibodies have received FDA or EMA EUA to treat COVID-19 either as cocktails of two antibodies or monotherapies (Table 2). Two of the cocktails, casirivimab plus imdevimab and bamlanivimab plus etesevimab do not neutralize SARS-CoV-2 Omicron variant. The third cocktail, cilgavimab plus tixagevimab, has a diminished neutralization potency when challenged with Omicron. Out of the three monotherapies, only sotrovimab and bebtelovimab preserved their neutralizing potency when challenged with SARS-CoV-2 Omicron, whereas regdanvimab lost it. Therefore, most of the FDA and/or EMA EUA antibodies have lost efficacy with the VOCs, in particular Omicron. IgG-A7 neutralizes WT, Delta and Omicron and hence, it seems to be a good lead candidate for developing broadly neutralizing anti-SARS-CoV-2 prophylactic and/or therapeutic drugs.

All the nine therapeutic antibodies except sotrovimab and bebtelovimab recognize epitopes in the RBD that do map into residues in contact with hACE2 (Figure 6, top). Sotrovimab is the only therapeutic antibody binding residues away from the RBD:hACE2 interface (Figure 6, bottom panel) and hence, it does not block RBD:hACE2 interaction. IgG-A7 blocks the RBD:hACE2 interaction [12] and thus seems to recognize residues in the RBD interface with hACE2, suggesting a mechanism of neutralization which is different from sotrovimab.

Bebtelovimab, on the other hand, mostly recognizes conserved residues in RBD WT, Delta and Omicron (Figure 6, middle panels). The contact residues are located in the periphery of the RBD interface with hACE2 and bebtelovimab mechanism of neutralization is by sterically blockade of the RBD interaction with hACE2 rather than binding residues in the core of the RBD interface with hACE2. This explains why bebtelovimab binds SARS-CoV-2 RBD WT, Delta and Omicron, despite the large number of mutations in Omicron with respect to the WT and Delta. Since IgG-A7 competes with the RBD for binding to hACD-2 and neutralizes Omicron, whereas all of the other therapeutic antibodies that bind residues in the RBD:hACE2 interface do not, it can be hypothesized that IgG-A7 might have a similar mechanism of neutralization than bebtelovimab, i.e., binding conserved residues in the periphery of the RBD:hACE-2 and sterically blockage of the RBD:hACE2 interaction. Consistent with this suggestion, bebtelovimab belongs to class III of SARS-CoV-2 neutralizing antibodies, which are encoded by the IGHV1 and IGHV2 gene families. As reported previously [12] IgG-A7 is encoded by the IGHV1-24 germline gene, allowing to include IgG-A7 in this class of SARS-CoV-2 neutralizing antibodies.

Finally, worth mentioning is that additional developability and efficacy studies are in progress in our laboratory using a larger number of K18-ACE2 mice, infected with higher doses of SARS-CoV-2 WT, Delta and Omicron. The results of additional developability assays indicate (manuscript in preparation) that IgG-A7 can be expressed in CHO cells with a yield to 200 mg/L after 14 days of culture, can be formulated at 40 mg/mL in a standard formation buffer and is stable for at least 6 months stored at 2–8 °C. The efficacy studies, on the other hand, is consistent with that reported in this work for a 5 mg/Kg dose but further show that IgG-A7 could protect 100% of the mice at a lower dose of 0.5 mg/Kg. Thus, although these results should further be confirmed, the broadly neutralization profile of IgG-A7, together with a potential low therapeutic dose, should have an impact into lowering manufacturing costs and perhaps enabling a subcutaneous administration. Therefore, IgG-A7 seems to be an excellent candidate for developing potent antibody-based drugs to prevent and/or treat patients infected with SASR-CoV-2 VOCs.

## 5. Patents

A PCT (Patent Cooperation Treaty) protecting the antibody sequences described in the work is in the process of being filed.

## Figures and Tables

**Figure 1 antibodies-11-00057-f001:**
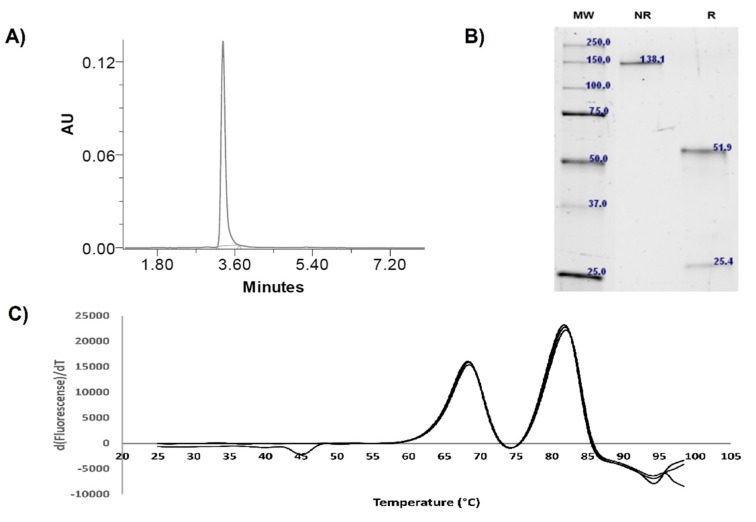
Physicochemical characterization of IgG-A7. (**A**) Analytical SEC shows close to 100% monomeric content. (**B**) SDS-PAGE under non-reducing conditions indicates a single band at 138 kDa and two bands: 51.9 kDa (heavy chain) and 25.4 kDa (light chain) under reducing conditions showing. (**C**) Protein Thermal Shift™ assay shows two unfolding transitions: 68.5 °C and 82.1 °C.

**Figure 2 antibodies-11-00057-f002:**
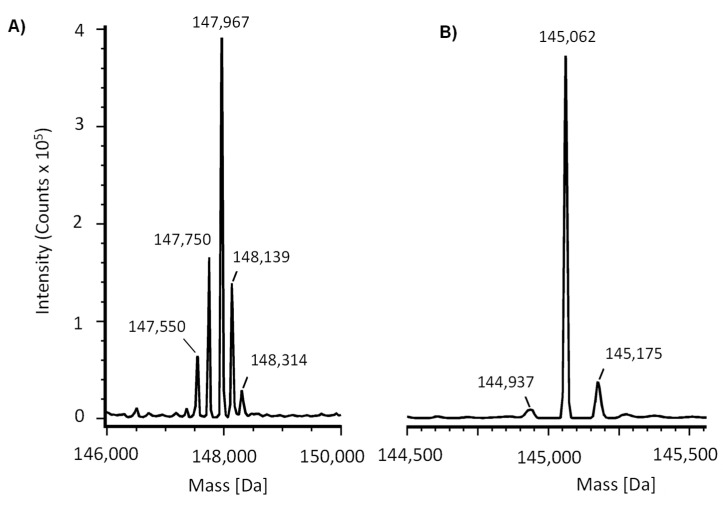
Mass Spectrometry analysis of IgG-A7, a broadly SARS-CoV-2 neutralizing antibody. Main isoforms of the intact (**A**) and deglycosylated (**B**) molecules.

**Figure 3 antibodies-11-00057-f003:**
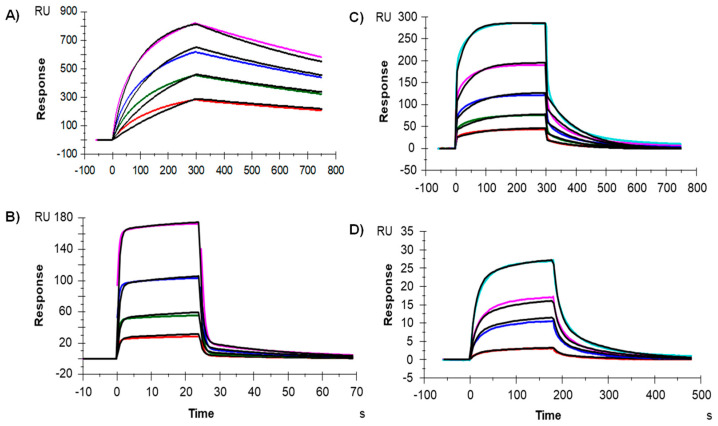
Binding of IgG-A7 to (**A**) FcγRIA, (**B**) FcγRIIA, (**C**) FcγRIIIA and (**D**) FcRn.

**Figure 4 antibodies-11-00057-f004:**
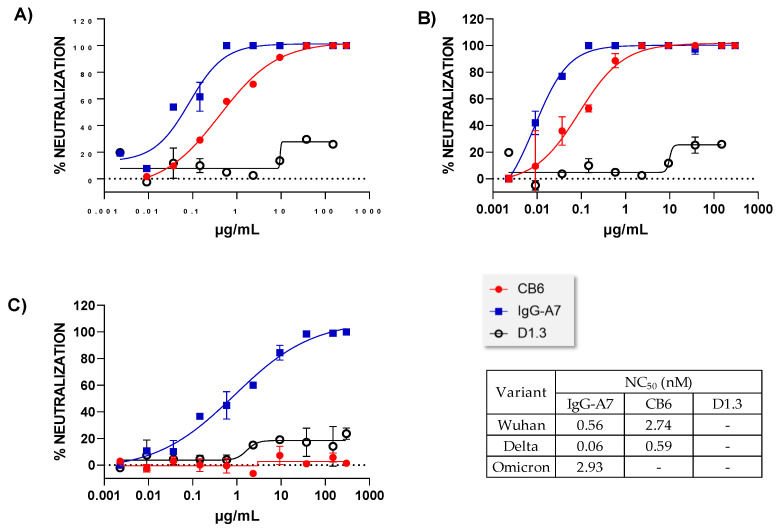
Dose-response neutralization curves of IgG-A7, CB6 and D1.3 (negative control) in PRNT for different variants of SARS-CoV-2. (**A**) WT, (**B**) Delta, and (**C**) Omicron. The Table on the left-bottom corner reports the NC_50_ values obtained by fitting the raw data to a four-parameter dose-response curve in GraphPad Prism 9.3.1.

**Figure 5 antibodies-11-00057-f005:**
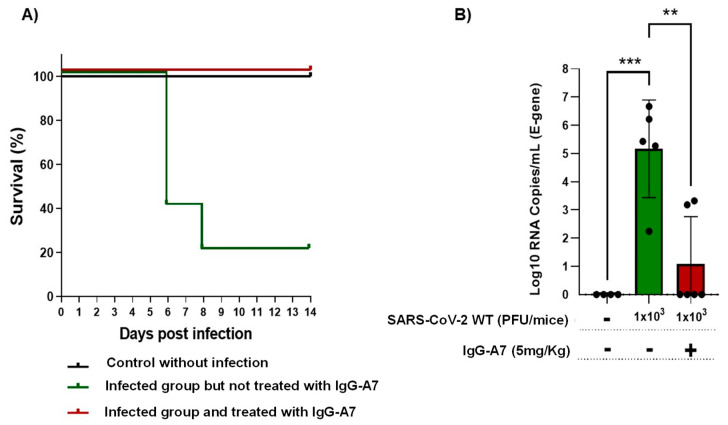
IgG-A7 protection of K18-ACE2 mice expressing hACE2 infected with SARS-CoV-2 WT. The efficacy of IgG-A7 was assessed through survival (**A**) and viral load (**B**) analyses. Statistics were performed through Kaplan Meier survival analysis (Chi^2^ = 11.30, *p* = 0.0035) while viral load was performed by one-way ANOVA (F2,12 = 16.27) with Dunnett’s post hoc test (** *p* < 0.01; *** *p* < 0.001).

**Figure 6 antibodies-11-00057-f006:**
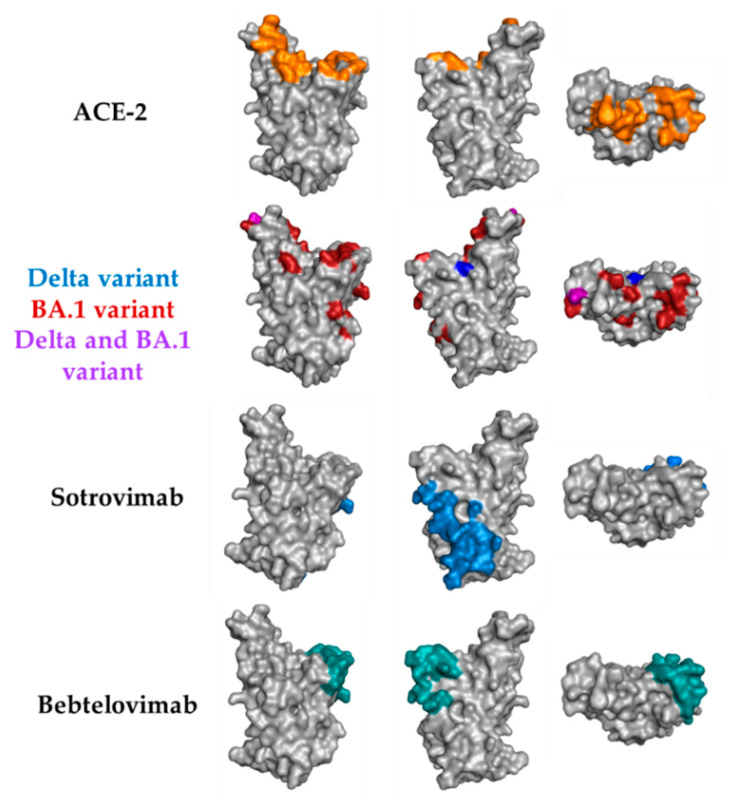
Connolly surface of the RBD side and top views mapping the residues involved in the interaction with hACE2 (top), mutations of the Delta and Omicron (BA.1) variants with respect to the RBD WT and epitopes of sotrovimab and bebtelovimab. The figures were generated with PyMOL Molecular Graphics System version 2.4.1 using the PDB ID: 6VW1 for the RBD:hACE2 interface and mutations of Delta and Omicron, PDB ID: 7SOC for the epitope of sotrovimab and PDB ID: 7MMO for the epitope of bebtelovimab.

**Table 2 antibodies-11-00057-t002:** Neutralization of SARS-CoV-2 by the nine EUA FDA and/or EMA antibodies. SARS-CoV-2 variant D614G was reported as early in the COVID-19 pandemic as January 2020. Omicron variant was reported almost two years later (November 2021).

(INN)	D614G	(Omicron)
Casirivimab plus Imdevimab	++++	−
Bamlanivimab plus Etesivimab	++++	−
Cilgavimab plus Tixagevimab	++++	Cilgavimab (++)
		Tixagevimab (+)
Regdanvimab	++++	−
Sotrovimab	++	++
Bebtelovimab	++++	++++

Data were taken and modified from Van Blargan et al. [33] and Westendorf et al. [34].

## Data Availability

All the data obtained during this study is included in the manuscript. Additional information could be provided by the authors upon a reasonable request.

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
