# Peer review of "In Vitro and In Vivo Characterization of a Broadly Neutralizing Anti-SARS-CoV-2 Antibody Isolated from a Semi-Immune Phage Display Library"

_2073-4468, 2022, doi:10.3390/antib11030057_

Round 1
Reviewer 1 Report
The manuscript “In vitro and in vivo characterization of a broadly neutralizing anti-SARS-CoV-2 antibody isolated from a semi-immune phage display library” by Edith González-González et al. adds to the growing number of reports describing monoclonal antibodies capable of neutralizing multiple variants of the coronavirus causing current Covid-19 epidemic.
At this point in the evolution of SARS -Cov-2, Omicron, and its subvariants are of greatest concern, and as much as broad neutralization activity is of scientific interest, the activity against Omicron is most important. Among the three variants tested, the efficacy of IgG-A7 against the Omicron variant was still the lowest, which suggests that a clinically applicable antibody may have to be an engineered version of IgG-A7 with a higher affinity. Nevertheless, the manuscript reports valuable findings of interest to the scientific community; it is clearly written, makes no highly speculative claims, and therefore it should be considered.
I have the following comments that I wish the authors to address in the next version of the manuscript:
Thorough developability assessment usually includes measuring pharmacokinetics in mice, determination of chemical liabilities (deamination, oxidation, aggregation propensity, etc.), and determining antibody solubility across a range of pH. The solubility profile would be especially important if the antibody could indeed be administered subcutaneously.
The assessment presented in the manuscript is of rather limited scope, although the authors seem to feel otherwise.
The extinction coefficient of 1 (mg/mL)cm-1 seems lower than the typical extinction coefficient value for the majority of IgG antibodies (1.3-1.5). Please explain if the value used is calculated based on the antibody sequence and, if not, why.
Y-axis is missing in figure 1 c.
Please specify the route of administration of the antibody IgG-A7 to mice.
Why did the authors decide not to evaluate the binding affinity of IgG-A7 to S1 protein derived from WT, Delta, and Omicron variants?
Authors should consider updating the manuscript with more literature references reporting the discovery/characterization of monoclonal antibodies with broad neutralizing activity and specifically neutralizing Omicron variants with high potency.
Author Response
Reviewer:
The manuscript “In vitro and in vivo characterization of a broadly neutralizing anti-SARS-CoV-2 antibody isolated from a semi-immune phage display library” by Edith González-González et al. adds to the growing number of reports describing monoclonal antibodies capable of neutralizing multiple variants of the coronavirus causing current Covid-19 epidemic.
At this point in the evolution of SARS -Cov-2, Omicron, and its subvariants are of greatest concern, and as much as broad neutralization activity is of scientific interest, the activity against Omicron is most important. Among the three variants tested, the efficacy of IgG-A7 against the Omicron variant was still the lowest, which suggests that a clinically applicable antibody may have to be an engineered version of IgG-A7 with a higher affinity. Nevertheless, the manuscript reports valuable findings of interest to the scientific community; it is clearly written, makes no highly speculative claims, and therefore it should be considered.
I have the following comments that I wish the authors to address in the next version of the manuscript:
Thorough developability assessment usually includes measuring pharmacokinetics in mice, determination of chemical liabilities (deamination, oxidation, aggregation propensity, etc.), and determining antibody solubility across a range of pH. The solubility profile would be especially important if the antibody could indeed be administered subcutaneously.
The assessment presented in the manuscript is of rather limited scope, although the authors seem to feel otherwise.
Response:
Thank you for your comments.
The tests suggested by the reviewer should certainly be performed as part of the antibody characterization and preparation for preclinical development - see also referee 2 comments for additional testing. Notice however that this manuscript is a continuation of the first paper published in Antibodies (https://doi.org/10.3390/antib11010013). In that report we described the isolation and preliminary testing of IgG-A7. To complement the results of the first publication and thus provide support to the broad neutralization profile of IgG-A7, we are submitted this second manuscript, which focuses on the in vitro and in vivo characterization of IgG-A7.
To ensure the assays were performed with a high-quality material we performed some critical developability tests that provided information on the purity, identity, stability, and binding of the molecule to diverse Fc gamma receptors, as well as FcRn binding. Having concluded that the antibody is well-behaved in these experiments, the emphasis of the manuscript is on the SARS-CoV-2 VOCs neutralization in vitro and efficacy.
Having said so and to avoid confusion, we removed the word “thoroughly” form the beginning of the discission and replaced “developability” with “physicochemical characterization”.
Worth mentioning is that we have performed additional studies as part of the development of IgG-A7. Some of the results such as additional efficacy studies are mentioned in the discussion of the manuscript. Other studies are not presented in the manuscript due to space limitations. These tests include expression of IgG-A7 in CHO cells and comparison with the material expressed in HEK293, peptide mapping using mass spectrometry, solubility, stability after five freeze/thaw cycles and formulation studies. We also performed the tests mentioned by the reviewer, in particular PTM modifications under stress conditions.
Studies in progress include long-term stability at diverse temperatures, performance of the antibody under stress conditions (as mentioned by the reviewer), tissue cross-reactivity in human and mouse to identify cross-reactive molecules/tissues, which can lead to toxicity and/or unwanted reactions in vivo.
All the above studies have been (or are) performed with material generated in CHO cells in preparation for the stable cell line development. Stability, solubility and PTM under stress conditions indicate that the antibody can be formulated in a standard formulation buffer and concentrated to at least to 50 mg/mL - which is above the typical concentration for IV of 40 mg/mL.
Further characterization indicates that IgG-A7 does not aggregate nor loses binding after five freeze/thaw cycles or stored at 4oC for least for 6 months. It does not lose binding nor aggregate after stress conditions such as shacking or exposure to oxidant conditions either.
Together with the key experiments described in the manuscript, it gives us confidence that IgG-A7 is a good candidate for therapeutic development but, much has to be done yet to request authorization for clinical studies, which will take time and a substantial number of resources. In our opinion the results of these additional tests are beyond the scope of the manuscript.
Reviewer:
The extinction coefficient of 1 (mg/mL)cm-1 seems lower than the typical extinction coefficient value for the majority of IgG antibodies (1.3-1.5). Please explain if the value used is calculated based on the antibody sequence and, if not, why.
Response:
We internally use a coefficient of extinction of 1.3 mg/mL but, the CRO that is producing the material for preclinical studies and eventually will produce the clinical material uses 1 mg/mL. The latter is a general coefficient of extension for proteins, and we decided to report our results using it. Notice that all the concentration calculations, including the controls, can rapidly be corrected by using 1.3 mg/mL, if necessary.
Reviewer:
Y-axis is missing in figure 1 c.
Response:
Included in the revised version of the manuscript.
Reviewer:
Please specify the route of administration of the antibody IgG-A7 to mice.
Response:
We used intraperitoneal administration. Added. Thank you. It was certainly missed in the Material and method section.
Reviewer:
Why did the authors decide not to evaluate the binding affinity of IgG-A7 to S1 protein derived from WT, Delta, and Omicron variants?
Response:
IgG-A7 was selected with RBD from a phage display library as reported before (https://doi.org/10.3390/antib11010013). We showed in that report that it is specific for RBD and binds with high affinity Wuhan, Delta ad Omicron in both ELISA and BIAcore. Further, we performed competition with hACD2 and other relevant molecules to ensure the antibody is specific for RBD. In the current publication we tested neutralization with the whole virus in vitro and report efficacy studies showing the specificity of the antibody. Thus, we consider that assessing binding to S1 was unnecessary.
Reviewer:
Authors should consider updating the manuscript with more literature references reporting the discovery/characterization of monoclonal antibodies with broad neutralizing activity and specifically neutralizing Omicron variants with high potency.
Response:
Added additional references including reference 5, which is a review on the nine FDA and/or EMA approved anti-SASR-CoV-2 therapeutic antibodies and neutralization potency - or lack of thereof - when challenged with VOCs. This review was written by some of the authors and accepted this week in IJMS.
Reviewer 2 Report
Please find below the comments/suggestions/queries to your manuscript:
Introduction:
- Line 39: Inappropriate referencing. Zhang et. al., did not declare COVID-19 as global pandemic, WHO did. World Health Organization. "WHO Director-General’s opening remarks at the media briefing on COVID-19-11 March 2020." (2020).
If you want to quote a paper, then this should be the one:
WHO Declares COVID-19 a Pandemic
Acta Biomed. 2020; 91(1): 157–160
10.23750/abm.v91i1.9397
- Line 53: Reference should be mentioned here. Authors can also mention the study published in The Lancet, published in early July 2022:
"Monoclonal antibody therapies against SARS-CoV-2"
https://doi.org/10.1016/S1473-3099(22)00311-5
- Line 75: in vitro not In vitro
Query: Why did the authors not consider using bebtelovimab as a control in all their experiments as it blocks RBD (Omicron)-hACE2 interaction and neutralizes omicron as the A7. It would have been an appropriate benchmark to compare A7. Instead they choose to use CB6, which is a precursor of etesivimab. An explanation is needed here.
Materials and Methods
2.1 mAb production and purification:
- Line 86: Any specific reason why transfected cells were incubated only 4 days and not longer (5-6 days)? What was the culture volume? This question crops from the fact that the reported yield of A7 is only 20 mg/L (which is on the lower end for IgG1 production using HEK293T cells). Did the authors test different HC:LC ratios to augment the production?
- Line 98/99: Content repetition! Those lines have been repeated from line 75/76/77.
2.2 Developability assessment:
- Line 118: Why was SEC performed at 30°C? Any specific reason for that? Was checking thermal stability the reason? Could the authors comment on extended thermal stability of this A7 IgG1 (for example how does A7 hold at 37°C for 2-3 weeks?)
2.3 MS characterization:
- Line 136: Expand the acronym FA as formic acid.
- Line 142: Expand the acronym AcN as acetonitrile.
2.4 Binding to Fc-gamma and FcRn by SPR:
- Line 169/170/171: Authors mention that recombinant FcRn was directly immobilized on a CM5 chip as previously described at 300 RUs but the reference is missing!!
- Line 177-182: Does this paragraph provide details about FcRn binding experiment? If yes, then it should be explicitly mentioned. If not, the authors need to provide this information.
2.5 SARS-CoV-2 isolates:
- Line 184/185: Frame the sentence correctly. Sentence ends abruptly.
- Line 188: Vero cells were incubated for 60 hours and then frozen for one day to lysate cells (lyse the cells).
- Line 194/195/196: Frame the sentence correctly. Grammatically not correct.
2.6 PRNT test:
- Line 214: Why was A7, CB6 and D1.3 incubated with Omicron for 2 hours and not 1 hour as was done with WT and Delta variants. Authors already had the information that both CB6 and D1.3 will not bind to RBD of Omicron. Why was additional time given to A7 with respect to Omicron binding? Was 1 hour incubation tested with A7?
2.7 Efficacy assay:
- Line 235: Viral load in the lung was determined for by RT-PCR to detect SARS-235 CoV-2 E gene copies on day 14 post-infection
Results
3.1 Expression and physicochemical characterization of IgG-A7:
- Line 248: Have the authors tested the Fab melting as well (apart from IgG melting)?
- Line 272: At what pH was FcRn binding of 2.74 uM reported?
In general, can the authors comment about the active fractions (based on RL value) of FcγRIA, FcγRIIA, FcγRIIIA and FcRn?
3.2 Neutralization assay in vitro:
- Line 280/281: IC50 for etesevimab is 40 pM to Delta variant (figure 1c of Wang et. al., 10.1080/22221751.2022.2032374) as compared to 590 pM reported for CB6 here. Authors should have chosen clinical molecules instead of precursor molecules for benchmarking.
3.3 Efficacy in transgenic mice:
- Why was % survival not measured/shown for CB6? Especially when authors chose to benchmark with CB6 for all other relevant assays.
- Was the A7-IgG administered in single dosing (24 h post-infection)?
- Data for only 5 mice is visible in viral load histogram (figure 5b) for untreated mice injected with WT SARS-CoV-2. Are any of the data overlapping/masking the 6th mouse data? Also, 2 mice in the A7 treated seem to be outlier. Can authors comment if those outliers show any gender bias?
Discussion
- Line 321/322: This is a gross extrapolation and authors should refrain from making such assumptions. Can the authors comment on the hACE2 receptors density in the transgenic mice (they used) and compare it with patient lung samples?
- Line 322/323/324: Authors compare their hypothetical/estimated therapeutic dosing of A7 with etesevimab (not precursor CB6) and then attribute this to higher in vitro neutralization potency of A7 over CB6. This disturbs and creates a logical fallacy.
- Line 353, 360, 380 and 382: Consider rephrasing. Grammatically not correct!
- Line 372-385: As mentioned arlier, this conclusion is much too far-fetched and should be avoided
Reviewer recommendations for additional experiments:
1. The story could have been more convincing if the authors would have reported any histology data of the lung tissue (or of any other relevant organ) post euthanization (and compare between non-treated and treated mice).
2. The publication would be even more robust if authors could compare the viral load with etesivimab and bebtelovimab.
3. Experiments could be performed for determining the exact binding foorprint of A7 on RBD of WT, Delta and Omicron variants instead of gross speculations with bebtelovimab binding.
4. If authors strongly suggest to take A7 as a candidate to treat or prevent Covid-19, then they definitely need to comment on the folowing aspects;
a. Test of inhibition of infection with A7 in monkeys in both prophylactic and treatment settings
b. Self-aggregation propensity of A7 using IgG-based AC-SINS
c. Poly-reactivity assessment of A7 IgG to DNA and Insulin (if not with BVPs)
d. Accelerated stability/forced degradation studies if the HCDRs or LCDRs have any sequence liabilities like oxidation/isomerization/deamidation etc. (cannot comment more as the sequences were not revealed here).
Author Response
First of all, thank you for the detailed review of the manuscript and useful comments to improve its quality. A point-by-point response follows.
Reviewer:
Introduction:
- Line 39: Inappropriate referencing. Zhang et. al., did not declare COVID-19 as global pandemic, WHO did. World Health Organization. "WHO Director-General’s opening remarks at the media briefing on COVID-19-11 March 2020." (2020).
If you want to quote a paper, then this should be the one:
WHO Declares COVID-19 a Pandemic
Acta Biomed. 2020; 91(1): 157–160
10.23750/abm.v91i1.9397
Response:
Misplaced the reference. Corrected.
Reviewer:
Line 53: Reference should be mentioned here. Authors can also mention the study published in The Lancet, published in early July 2022:
"Monoclonal antibody therapies against SARS-CoV-2"
https://doi.org/10.1016/S1473-3099(22)00311-5
Response:
Added suggested references. Also, included reference 5, which is a review on FDA and/or EMA approved anti-SASR-CoV-2 therapeutic antibodies and neutralization potency -or lack of thereof - when challenged with VOCs. This review was written by some of the authors and accepted this week in IJMS.
Reviewer:
- Line 75: in vitro not In vitro
Query: Why did the authors not consider using bebtelovimab as a control in all their experiments as it blocks RBD (Omicron)-hACE2 interaction and neutralizes omicron as the A7. It would have been an appropriate benchmark to compare A7. Instead they choose to use CB6, which is a precursor of etesivimab. An explanation is needed here.
Response:
Bebtelovimab would certainly be a good comparator. However, please notice that we started development of IgG-A2 (https://doi.org/10.3390/antib11010013) right after Omicron emerged in November 2021. At that point in time not much information on the neutralization profile of the approved therapeutic antibodies was available.
We have been using CB6 as control antibody during the development of IgG-A7 and other anti-SARS-CoV-2 antibodies we have isolated and optimized. We have cloned and produced CB6 with the same isotype than IgG-A7, and express and purify it in the same conditions we use for the other antibodies. Therefore, we consider CB6 as a good positive control and a proper reference to validate our assays.
Reviewer:
Materials and Methods
2.1 mAb production and purification:
- Line 86: Any specific reason why transfected cells were incubated only 4 days and not longer (5-6 days)? What was the culture volume? This question crops from the fact that the reported yield of A7 is only 20 mg/L (which is on the lower end for IgG1 production using HEK293T cells). Did the authors test different HC:LC ratios to augment the production?
Response:
Good observation. We express our antibodies for four days because it is a good compromise between time and expression yield – it gives us enough material for testing. If we find issues with a given antibody, we can quickly move to characterize other antibodies and hence, save time and resources. Knowing that four days is a short time for expressing an antibody, we explicitly mentioned it in Material and method.
It should be mentioned though that we have tested expression of IgG-A7 in different volumes and different expression platforms including HEK293, Expi and CHO. In CHO we have expressed IgG-A7 at 30 mL and 1L scales more than once, with incubation times of 14 days. IgG-A7 consistently expresses at 200 mg/L in CHO after 14 days. We added a line in the discussion of the manuscript to complement the comment of expression yield in HEK293.
Reviewer:
- Line 98/99: Content repetition! Those lines have been repeated from line 75/76/77.
Response:
Fixed.
Reviewer:
2.2 Developability assessment:
- Line 118: Why was SEC performed at 30°C? Any specific reason for that? Was checking thermal stability the reason? Could the authors comment on extended thermal stability of this A7 IgG1 (for example how does A7 hold at 37°C for 2-3 weeks?)
Response:
Typical SEC experiments are performed between 10oC and 30oC (see: https://www.agilent.com/cs/library/brochures/br-ordering-guide-sec-mab-fragments-dimers-5994-3947en-agilent.pdf). We have chosen to use 30oC. As explained in the reference, high temperatures improve the resolution and recovery of hydrophobic peptides and hence, we can better detect aggregates in the preparation.
Long-term (8 weeks) stability test at 37oC is in progress. We have been running binding experiments with preparations of the antibody produced 6 months ago, which we store at 4-8oC. We have also conducted stability studies (freeze/thaw cycles) in PBS and formulation buffer. The antibody does not aggregate and retains close to 100% binding after 6 months of storage at 4-8oC, as well as after five freeze/thaw cycles. See also comments to referee 1.
Reviewer:
2.3 MS characterization:
Reviewer:
- Line 136: Expand the acronym FA as formic acid.
Response:
Done.
Reviewer:
- Line 142: Expand the acronym AcN as acetonitrile.
Response:
Done.
Reviewer:
2.4 Binding to Fc-gamma and FcRn by SPR:
Reviewer:
- Line 169/170/171: Authors mention that recombinant FcRn was directly immobilized on a CM5 chip as previously described at 300 RUs but the reference is missing!!
Response:
Modified this section and added references on FcRn and Fc gamma receptors in Table 1.
Reviewer:
- Line 177-182: Does this paragraph provide details about FcRn binding experiment? If yes, then it should be explicitly mentioned. If not, the authors need to provide this information.
Response:
Modified this section and added references.
Reviewer:
2.5 SARS-CoV-2 isolates:
- Line 184/185: Frame the sentence correctly. Sentence ends abruptly.
Response:
Corrected.
- Line 188: Vero cells were incubated for 60 hours and then frozen for one day to lysate cells (lyse the cells).
Response:
Corrected.
Reviewer:
- Line 194/195/196: Frame the sentence correctly. Grammatically not correct.
Response:
Corrected.
Reviewer:
2.6 PRNT test:
- Line 214: Why was A7, CB6 and D1.3 incubated with Omicron for 2 hours and not 1 hour as was done with WT and Delta variants. Authors already had the information that both CB6 and D1.3 will not bind to RBD of Omicron. Why was additional time given to A7 with respect to Omicron binding? Was 1 hour incubation tested with A7?
Response:
We certainly tried 1 hour incubation and saw an attenuated signal. Thus, we increased the incubation time to 2 hours and that is the result we are reporting.
Running CB6 and D1.3 side-by-side with IgG-A7 in the Omicron experiment was useful as negative control to discard false positives in the experiment, matrix interference and non-specific binding.
Reviewer:
2.7 Efficacy assay:
- Line 235: Viral load in the lung was determined for by RT-PCR to detect SARS-235 CoV-2 E gene copies on day 14 post-infection
Response:
Corrected.
Reviewer:
Results
3.1 Expression and physicochemical characterization of IgG-A7:
- Line 248: Have the authors tested the Fab melting as well (apart from IgG melting)?
Response:
We haven’t tested the Fab. We commonly test the thermal stability of the whole molecule in the final therapeutic format (IgG1). It is known that the CH2 unfolds around 68oC. Other unfolding transitions should correspond to the Fab and/or CH3 domain (see: DOI: 10.1007/978-1-61779-520-6_14).
Reviewer:
- Line 272: At what pH was FcRn binding of 2.74 uM reported?
Response:
The evaluation was performed at pH 6.0 – now added. We also added the temperature (25oC), which was missed. Thank you for identifying this omission as it is critical to understand the FcRn binding assay.
Reviewer:
In general, can the authors comment about the active fractions (based on RL value) of FcγRIA, FcγRIIA, FcγRIIIA and FcRn?
Response:
I guess the reviewer meant “RU” (Resonance Units)”. Not sure what the reviewer meant by active fractions.
Reviewer:
3.2 Neutralization assay in vitro:
- Line 280/281: IC50 for etesevimab is 40 pM to Delta variant (figure 1c of Wang et. al., 10.1080/22221751.2022.2032374) as compared to 590 pM reported for CB6 here. Authors should have chosen clinical molecules instead of precursor molecules for benchmarking.
Response:
We report 2.74 nM for CB6 and SARS-CoV-2 Wuhan and 0.59 nM for CB6 and Delta (see Table imbedded in Figure 4).
Figure 1c of Wang et al shows neutralization data generated with a pseudovirus assay. We used an authentic virus in the neutralization assay. In Wang et al paper, p. 550, left column, close to the end of the first paragraph, Wang et al state “Individual Nabs, etesevimab or JS026, exhibited ND50 of 256 or 86 ng/mL to authentic SARS-CoV-2 virus”. 256 ng/mL = 1.71 nM. This value is very close to our report (2.74 nM) - less than a two-fold difference.
Regarding the comment of using clinical molecules instead of precursors, as explained above, we have been using CB6 as a control antibody to validate our assays during the development of IgG-A7 and other anti-SARS-CoV-2 antibodies we have isolated and optimized. In our opinion, the focus of this paper is on reporting in vivo and in vitro characterization IgG-A7 using validated assays with a reference molecule. Since CB6 is a well-known antibody and plenty of information on its characterization is available, we provided a comparison with CB6 to give a reader a reference so s/he can assess the performance of IgG-A7. The focus on the paper is on IgG-A7 not a comparison with clinical molecules.
Reviewer:
3.3 Efficacy in transgenic mice:
- Why was % survival not measured/shown for CB6? Especially when authors chose to benchmark with CB6 for all other relevant assays.
Response:
Running side-by-side CB6 (or any other antibody) in efficacy studies will increase the number of experiments exponentially. If we factor in that we must run the efficacy tests with three SARS-CoV-2 strains (WT, Delta and Omicron) - as we have done already - it will result in a very large number of experiments.
As the referee surely is aware of, efficacy studies with SARS-CoV-2 are expensive as the experiments must be done under strict security protocols to avoid contaminations with the virus. Not to mention the costs in animals and reagents, plus justifying to and getting approval from ethical committees to increase the number of animals in the experiments.
All of the above, compounded with that, in our opinion, testing the efficacy of other antibodies won’t add much to the conclusion that IgG-A7 protects 100% of the transgenic mice infected with SARS-CoV-2, discouraged us from adding CB6 (or any other antibody) in the current efficacy studies.
Reviewer:
- Was the A7-IgG administered in single dosing (24 h post-infection)?
Response:
We mentioned that the treatment was applied next day after the infection but agree, stating that the treatment was applied 24 hours post-infection is a more accurate description of the procedure. We added a line stating that the treatment was applied 24 hours post-infection.
Reviewer:
- Data for only 5 mice is visible in viral load histogram (figure 5b) for untreated mice injected with WT SARS-CoV-2. Are any of the data overlapping/masking the 6th mouse data? Also, 2 mice in the A7 treated seem to be outlier. Can authors comment if those outliers show any gender bias?
Response:
Thank you for noticing it. This group has only 5 mice – now clarified in the text of the manuscript.
The two mice with a higher viral burden in the lungs in the treated group are male. We have consistently observed a gender bias in our experiments. Females are more resistant to the infection and respond better to the treatment.
Reviewer:
Discussion
- Line 321/322: This is a gross extrapolation and authors should refrain from making such assumptions. Can the authors comment on the hACE2 receptors density in the transgenic mice (they used) and compare it with patient lung samples?
Response:
Fully agree with the reviewer that a direct extrapolation of efficacy in mice to humans is a gross approximation. In fact, we alerted the reader in the discussion saying that “it is a somewhat simplistic calculation”. Nevertheless, with this extrapolation we wanted to provide a sense of the potential efficacy of IgG-A7 in humans. Notice that this kind of gross assumptions are a common place as starting point for determining the therapeutic dose in humans – that is why we use animals as preamble to clinical studies.
Additional considerations based on safety studies in phase I are necessary to scale up the doses in phase II and stablish the final therapeutic dose. Needless to say, we won’t know the safety and efficacy of IgG-A7 profile in humans until we complete the clinical trials. In the meantime, we need to make assumptions based on the data we can gather in animal models to decide on whether we want to spend millions in cell-line development and clinical testing.
The mice we use were purchased at Jackson Laboratories as described in Material and method. These mice are well characterized and frequently used in the efficacy studies of anti-SARS-CoV2 antibodies (see: https://www.nature.com/articles/s41590-020-0778-2).
Reviewer:
- Line 322/323/324: Authors compare their hypothetical/estimated therapeutic dosing of A7 with etesevimab (not precursor CB6) and then attribute this to higher in vitro neutralization potency of A7 over CB6. This disturbs and creates a logical fallacy.
- Line 353, 360, 380 and 382: Consider rephrasing. Grammatically not correct!
Response:
Corrected
Reviewer:
- Line 372-385: As mentioned arlier, this conclusion is much too far-fetched and should be avoided
Response:
Corrected
Reviewer:
Reviewer recommendations for additional experiments:
- The story could have been more convincing if the authors would have reported any histology data of the lung tissue (or of any other relevant organ) post euthanization (and compare between non-treated and treated mice).
Response:
Agree with the referee that histology data can further support the results of the efficacy study and we are considering it as we have collected the tissues of the treated and untreated mice. However, in our opinion, the main result of this experiment is the survival curves. The viral burden in the lungs is just supporting data showing that the survival correlates with the viral burden in the lungs.
Reviewer:
- The publication would be even more robust if authors could compare the viral load with etesivimab and bebtelovimab.
Response:
Agree but, in our opinion, beyond the scope of the manuscript.
Reviewer:
- Experiments could be performed for determining the exact binding foorprint of A7 on RBD of WT, Delta and Omicron variants instead of gross speculations with bebtelovimab binding.
Response:
Agree. Epitope mapping, in particular by x-ray crystallography or high-resolution electron microscopy, are the golden methods to identify the exact location of the region on RBD binding IgG-A7 and map the residues involved in the interaction between the binding molecules. However, notice that at this stage of the research we are just providing a testable hypothesis on possible mechanisms of neutralization of IgG-A7.
We mention this hypothesis in the discussion of the manuscript and certainly do not consider this a “gross speculation”. We have published results (https://doi.org/10.3390/antib11010013) indicating that (1) IgG-A7 competes with the RBD for hACE2 binding, (2) IgG-A7 competes with another antibody, in which we have determined the epitope, and (3) we have shown that IgG-A7 binds all the three RBD variants, which differs by a large number of known mutations in the RBD. The latter is equivalent to a mutagenesis study.
All these facts together provide us with an informed guess on the possible location of IgG-A7 epitope on the RBD. Notice that this hypothesis is part of the discussion of the manuscript where we suggest a possible mechanism of neutralization. In our opinion, testable hypothesis as part of the discussion of a manuscript, based on information described in the results, are valid and help the reader to contextualize the results and scope of the study, while setting the stage for further research.
Reviewer:
- If authors strongly suggest to take A7 as a candidate to treat or prevent Covid-19, then they definitely need to comment on the folowing aspects;
a. Test of inhibition of infection with A7 in monkeys in both prophylactic and treatment settings
b. Self-aggregation propensity of A7 using IgG-based AC-SINS
c. Poly-reactivity assessment of A7 IgG to DNA and Insulin (if not with BVPs)
d. Accelerated stability/forced degradation studies if the HCDRs or LCDRs have any sequence liabilities like oxidation/isomerization/deamidation etc. (cannot comment more as the sequences were not revealed here).
Response:
We appreciate the reviewer’s feedback on the further steps to be taken during the preclinical development of an antibody. All these are certainly required tests and should be performed to request authorization for clinical trials, and we are in the process of performing these and additional tests – see also comments to referee 1 - as we prepare the dossier to request approval for clinical development. The results will eventually be reported elsewhere.
Round 2
Reviewer 2 Report
Reviewer:
In general, can the authors comment about the active fractions (based on RL value) of FcγRIA, FcγRIIA, FcγRIIIA and FcRn?
Response:
I guess the reviewer meant “RU” (Resonance Units)”. Not sure what the reviewer meant by active fractions.
Reviewer:
RLigand = Rmax x MWLigand/Stoichiometry x MWAnalyte
These are also reported in RUs. But upon comparison between theoretical and experimental values, we can calculate the % active fraction of the ligand in your preparation. Can you comment on that? - thank you